# Criticality and partial synchronization analysis in Wilson-Cowan and Jansen-Rit neural mass models

**Sheida Kazemi[1], AmirAli Farokhniaee[2], Yousef Jamali[1]***

**1** Biomathematics Laboratory, Department of Applied Mathematics, School of Mathematical Sciences, Tarbiat Modares University, Tehran, Iran, **2** School of Electrical and Electronic Engineering, University College Dublin, Dublin, Ireland

\* y.jamali@modares.ac.ir

**Data Availability Statement:** All relevant data are within the manuscript.

**Funding:** The author(s) received no specific funding for this work.

## Abstract

Synchronization is a phenomenon observed in neuronal networks involved in diverse brain activities. Neural mass models such as Wilson-Cowan (WC) and Jansen-Rit (JR) manifest synchronized states. Despite extensive research on these models over the past several decades, their potential of manifesting second-order phase transitions (SOPT) and criticality has not been sufficiently acknowledged. In this study, two networks of coupled WC and JR nodes with small-world topologies were constructed and Kuramoto order parameter (KOP) was used to quantify the amount of synchronization. In addition, we investigated the presence of SOPT using the synchronization coefficient of variation. Both networks reached high synchrony by changing the coupling weight between their nodes. Moreover, they exhibited abrupt changes in the synchronization at certain values of the control parameter not necessarily related to a phase transition. While SOPT was observed only in JR model, neither WC nor JR model showed power-law behavior. Our study further investigated the global synchronization phenomenon that is known to exist in pathological brain states, such as seizure. JR model showed global synchronization, while WC model seemed to be more suitable in producing partially synchronized patterns.

## 1 Introduction

The dynamics of macroscopic and mesoscopic brain activity is a challenging topic. The large-scale model activities can be complex and include a range of behaviors such as spiking and oscillatory, resting-state, chaotic, and periodic or nonperiodic activities [1, 2]. The neural mass models which are based on the mesoscopic scale, describe the action of neural populations rather than the behavior of spiking neurons and have been used broadly in modeling some brain activities such as epilepsy [3], sleep [4], and human alpha rhythm ($\sim$ 8- $\sim$ 12 Hz) [5]. These models depending on their output (voltage or rate) are classified into two voltage- and activity-based models [6]. The first voltage-based model in respect of one excitatory and one inhibitory ensemble was built by Lopes da Silva [7]. As an extension of Lopes da Silva's model, Zetterberg and colleagues included two excitatory and one inhibitory ensembles in their

**Competing interests:** The authors have declared that no competing interests exist.

description of the cortex, an approach that Jansen and Rits adopted for the definition of their model [8, 9]. The activity-based models are also referred to as Wilson-Cowan (WC) equations [10], Amari equations [11], Cohen–Grossberg equations [12], and neural field equations [13]. These types of models exhibit a vast range of interesting and important dynamics such as Hopf bifurcation, which motivated researchers to establish their reduced version counterparts useful for dynamical systems studies [14, 15].

The importance of synchronization in biological neuronal systems has been highlighted by numerous studies, demonstrating its critical role in brain functions. Neural activity synchronization is crucial to neural function and cognitive processes [16–19]. Synchronization of different single neuron classes to periodic external stimuli [20, 21] and coupled neurons [22–26] has been analyzed extensively, and can be measured both locally or over a global scale [27–30]. Brain networks are not the only complex systems that exhibit this phenomenon, social communications and financial markets are also affected. Bao and Huang conducted a large-scale natural field experiment to investigate gender differences in reaction to enforcement mechanisms. They explored the impact of external factors on individual behavior, which may have implications for the synchronization of neuronal networks [31]. Additionally, some analyses of market connectivity and risk spillover in stock prices by Chen et al. [32–34] may provide insight into the interconnectedness of neural networks and their synchronization. The management of epidemics is another situation that requires an understanding of the motivations and behaviors that lead to the dynamics and synchronization of complex networks [35–38]. Some practical example on how the mean field applications in different fields can be explored by analyzing their synchronization are presented in [2, 39–41].

Synchronization, desynchronization, and partial synchronization are three cases of dynamical behavior in systems of coupled oscillators that manifest interesting and neurophysiologically relevant results. In previous studies of neural synchronization, most of the attention has been focused on global synchronization, which means how all nodes in a network behave in unison [42, 43]. However, the experimental results do not exhibit this property in a healthy brain. Some severe brain disorders, such as epilepsy, Parkinson's, and essential tremor, may result from pathological synchronization [44–46]. However, partial synchronization within brain regions has been manifested for example in the macaque monkey's visual system experimentally [47].

Power-law distributions are crucial not only in neuronal systems but also in other vital apparatuses for living organisms. Scientists have been trying to observe power-law fluctuations in different temporal data and investigating their origins. By increasing the volume of this data, we can gain a better understanding of complex systems through empirical observations. Power-law behavior in various variables appears in critical dynamics. For example, the size and duration distributions of neuronal avalanches [48, 49] and EEG cascade [50, 51] obey a power-law. Moreover, long-range temporal correlations have been revealed in the amplitude envelopes of neural oscillations [52, 53]. The conceptual appeal of the power law distribution indicates that the system is operating near a critical point and is poising in a boundary between different types of dynamics that networks show the optimal performance of a cortical function [48, 54–56].

The following study compares the dynamics of WC firing rate model and the voltage-based Jansen-Rit (JR) model. Historically, WC equations are regarded as a fundamental and classic model in computational neuroscience. According to predictions, the WC equations will still have great utility for several decades to come [57]. JR is a minimal computational model of a small region of the cortex that belongs to a large group of models that are mathematically manifested by second-order linear differential equations. We simulated these two different types of neural mass models and investigated phase transition by taking into account the amount of

synchronization as an order parameter. Then, their partial synchronization is discussed in the following sections. Our results showed that the coexistence between synchronization and desynchronization occurred simultaneously only in the WC model. Moreover, although we also observed continuous phase transition in JR model, neither of these models showed power-law behavior.

## 2 Methods

### 2.1 Wilson-Cowan model

The classical WC model describes the dynamics of firing rates among neural populations in the brain [10, 58]. The activity of each neural population is computed as the mean firing rate of its excitatory ($E$) and inhibitory ($I$) subpopulations by using mean-field approximation [59]. The temporal evolution of the excitatory and inhibitory firing rates, $E(t)$ and $I(t)$, respectively, is governed by the following differential equations [10]:

$$\tau_E \frac{d}{dt} E(t) = -E(t) + \mathcal{S}[a_E(c_{EE}E(t) - c_{EI}I(t) - \theta_E + P(t))]$$
$$\tau_I \frac{d}{dt} I(t) = -I(t) + \mathcal{S}[a_I(c_{IE}E(t) - c_{II}I(t) - \theta_I)]$$

(1)

where $\tau_E$ ($\tau_I$) shows the time constant for excitatory (inhibitory) populations. The sigmoid function $\mathcal{S}$ introduces the thresholds $\theta_E$ and $\theta_I$ corresponding to the maximum slope values and can be different for excitatory and inhibitory subpopulations. Moreover, the slopes of the sigmoids are given by $a_E$ and $a_I$. $\mathcal{S}$ is given as follows:

$$\mathcal{S}(x) = \frac{1}{1 + e^{-x}}.$$

(2)

The synaptic weights are determined by the connectivity coefficients $c_{EE}$, $c_{EI}$, $c_{IE}$, and $c_{II}$. $P(t)$ represents the external stimuli acting on excitatory subpopulation in time t and is chosen randomly from a Gaussian distribution. Depending on parameter settings, specifically on the selection of the external input P(t), the dynamics of this dynamical system ranges from fixed-point relaxations to limit cycle oscillations. Table 1 shows the values of parameters with their interpretations. These values can produce oscillatory behavior in the WC model. An excitatory population interacts with an inhibitory population in each node (Fig 1(B)). A WC neural population network consists of nodes that correspond to distinct subcortical regions, as well as

**Table 1. Parameters in the WC model that shows the oscillatory behavior used in this study.**

| Parameters | Interpretation | Value |
|---|---|---|
| $\tau_E$ | Time constant for excitatory populations | 0.125 |
| $\tau_I$ | Time constant for inhibitory populations | 0.25 |
| $\theta_E$ | Maximum slope of sigmoid function for excitatory subpopulations | 2 |
| $\theta_I$ | Maximum slope of sigmoid function for inhibitory subpopulations | 8 |
| $a_E$ | The proportion of excitatory cells firing | 0.8 |
| $a_I$ | The proportion of inhibitory cells firing | 0.8 |
| $c_{EE}$ | Excitatory self-feedback synaptic weight | 8 |
| $c_{EI}$ | Synaptic weight from excitatory to inhibitory ensemble | 16 |
| $c_{IE}$ | Synaptic weight from inhibitory to excitatory ensemble | 8 |
| $c_{II}$ | Inhibitory self-feedback synaptic weight | 0.4 |

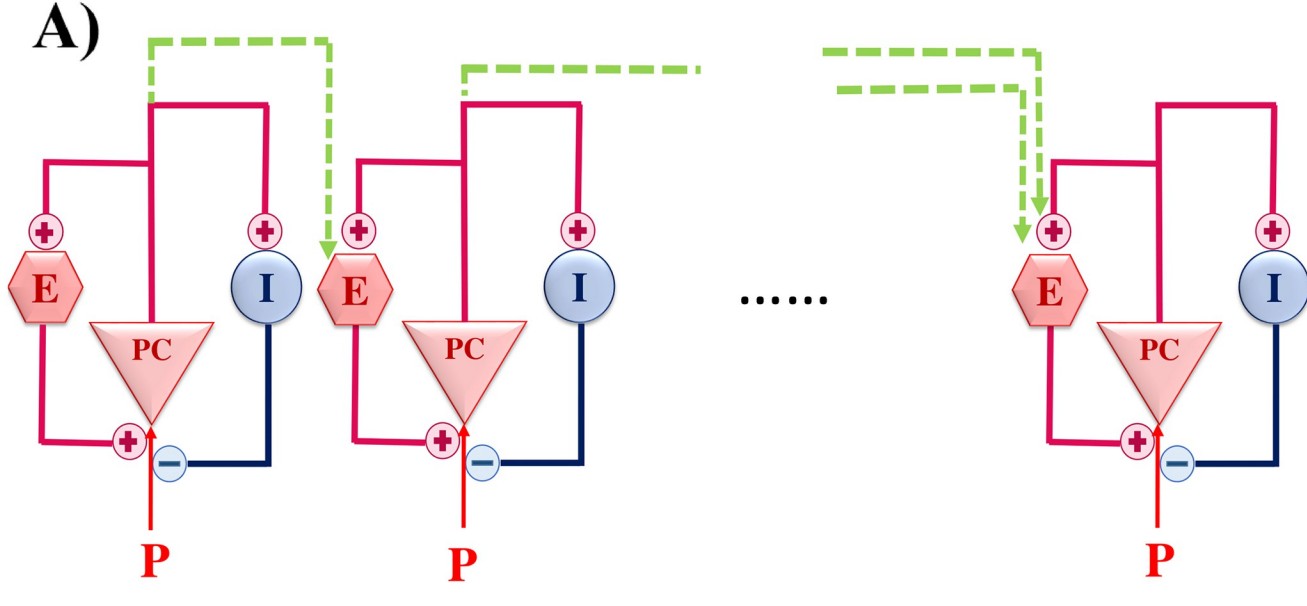

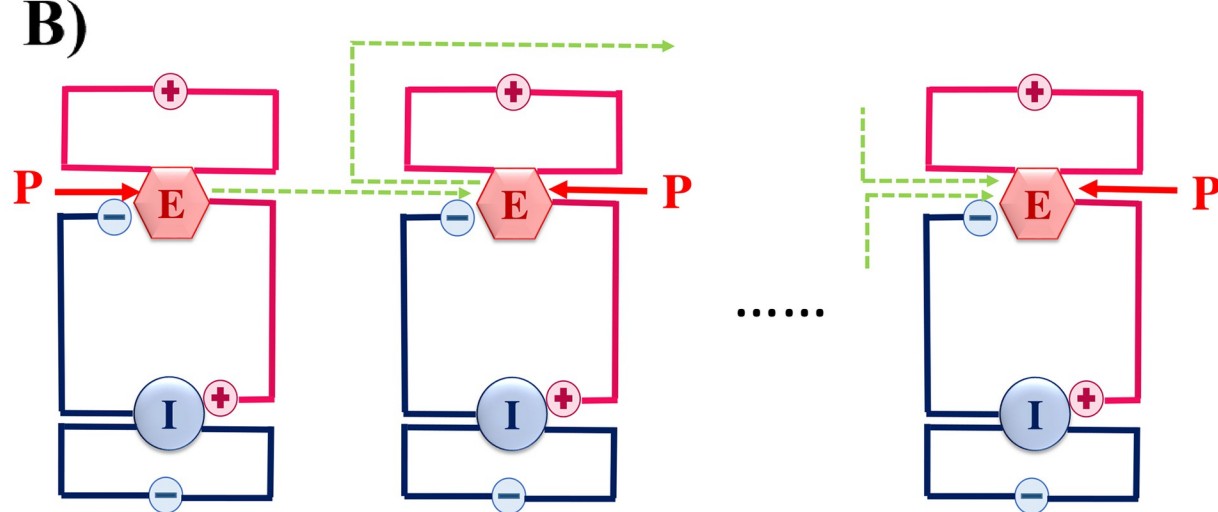

**Fig 1. Schematic representation of JR (A) and WC (B).** (A) A pyramidal cell (PC) with two excitatory and inhibitory interneurons in (**A**) interact with each other. (B) Two ensembles of excitatory and inhibitory neurons. P shows the external input. The green lines correspond to the connections between different regions or nodes in a network.

edges that represent structural connections between them. In a network, these nodes were linked through their connections only with excitatory populations.

In a network with N nodes, i = 1,. . ., N, the WC equations are as follows [60]:

$$
\tau_E \frac{d}{dt}E_i(t) = -E_i(t) + \mathcal{S}\left[a_E(c_{EE}E_i(t) - c_{EI}I_i(t) - \theta_E + P_i(t) + \mathcal{K}\sum_{l=1}^{N} M_{il}E_l(t)\right]
$$
$$
\tau_I \frac{d}{dt}I_i(t) = -I_i(t) + \mathcal{S}[a_I(c_{IE}E_i(t) - c_{II}I_i(t) - \theta_I)]
$$

(3)

where $M_{il}$ shows the network adjacency matrix and $\mathcal{K}$ represents coupling coefficient between nodes.

The appropriate choice of time constants $\tau_E$, $\tau_I$, and external inputs P(t) provided self-sustaining oscillations in the delta band frequency to match with the activity observed in [61].

## 2.2 Jansen-Rit model

JR model is based on the work of Lopes Da Silva [7]. A mathematical framework inspired by biology was developed to simulate spontaneous electrical activities of cortical columns, with an emphasis on alpha activity [8, 9]. In this model, pyramidal neurons receive input from excitatory and inhibitory interneurons in the same column as well as external input from other columns. This model can be written with six first-order differential equations as follows [8]:

$$
\begin{aligned}
\dot{y}_0(t) &= y_3(t) \\
\dot{y}_3(t) &= Aa\mathcal{S}(y_1(t) - y_2(t)) - 2ay_3(t) - a^2 y_0(t) \\
\dot{y}_1(t) &= y_4(t) \\
\dot{y}_4(t) &= Aa\{p(t) + C_2\mathcal{S}[C_1 y_0(t)]\} - 2ay_4(t) - a^2 y_1(t) \\
\dot{y}_2(t) &= y_5(t) \\
\dot{y}_5(t) &= BbC_4\mathcal{S}(C_3 y_0) - 2by_5(t) - b^2 y_2(t)
\end{aligned}
\tag{4}
$$

where $y_i$ for $i \in \{0, 1, 2\}$ represents the mean postsynaptic potentials of three neuronal populations ($y_0$ for pyramidal, $y_1$ for excitatory and $y_2$ for inhibitory neurons). Their deviations are denoted by $y_3$, $y_4$, and $y_5$, respectively. An external input is defined by the function $p(t)$, which may be generated from external sources or by neighbouring neural populations and is chosen randomly from a Gaussian distribution. $\mathcal{S}$ is a sigmoidal function that transforms the average membrane potential of neurons into the mean firing rate of action potentials and is given by:

$$
\mathcal{S}(v) = \frac{v_{max}}{1 + e^{r(v_0 - v)}}
\tag{5}
$$

where $v_{max}$ is the maximum firing rate of the neurons ensemble, $v_0$ is potential at half of the maximum firing rate and $r$ is the slope of the sigmoid at $v_0$. In Table 2, all of the parameters are quantified. According to Van Rotterdam's work, the parameters A, B, a, and b have been determined so that alpha activity can be produced by the system while respecting some basic properties of real post-synaptic potentials [62]. $v_0$, $v_{max}$, and r = $0.56 mV^{-1}$ are derived from the experimental studies of Freeman [63]. $C_i(i = 1, \ldots 4)$ are proportional to the average number of

**Table 2. Parameters in the JR model used in this study that are obtained experimentally.**

| Parameters | Interpretation | Value |
| --- | --- | --- |
| $A$ | Excitatory PSP amplitude | $3.25 mV$ |
| $B$ | Inhibitory PSP amplitude | $22 mV$ |
| $1/a$ | Time constant of excitatory PSP | $a = 100 s^{-1}$ |
| $1/b$ | Time constant of inhibitory PSP | $b = 50 s^{-1}$ |
| $C1, C2$ | Average numbers of synapses between excitatory populations | $1 * C, 0.8 * C$ |
| $C3, C4$ | Average numbers of synapses between inhibitory populations | $0.25 * C$ |
| $C$ | Average numbers of synapses between neural populations | $135$ |
| $v_{max}$ | Maximum firing rate | $5 Hz$ |
| $v_0$ | Potential at half of maximum firing rate | $6 mV$ |
| $r$ | Slope of sigmoid function at $v_0$ | $0.56 mV^{-1}$ |

synapses between populations. Jansen and Rit reduced these quantities to a single parameter C based on a neuroanatomical study [64] that used synaptic counting to estimate these quantities:

$$C_1 = C$$
$$C_2 = 0.8C$$
$$C_3 = 0.25C$$
$$C_4 = 0.25C$$

Using these values, Jansen and Rit varied C to observe alpha-like activity and obtained it for C = 135 [9].

The output of this model is $y_1 - y_2$, which represents the postsynaptic membrane potential of pyramidal neurons. The relationship between the incoming firing rate of pyramidal cells and the EEG signal is closely tied to the variable $y_1 - y_2$. The apical dendrites of pyramidal neurons are responsible for transmitting their postsynaptic potentials to the superficial layer of the cortex, which significantly contributes to the EEG signal [65]. In order to construct a network of interacting neural populations, representing a parcellation of the cerebral cortex, we consider a set of nodes and their links. Each area (node) can be represented by a neural mass model whose connections between them (edges) are based on structural connectivity.

In a network with N nodes, i = 1,..., N, the JR equations are described by the following set of ordinary differential equations [66]:

$$\dot{y}_{0_i}(t) = y_{3_i}(t)$$
$$\dot{y}_{3_i}(t) = Aa\mathcal{S}(y_{1_i}(t) - y_{2_i}(t)) - 2ay_{3_i}(t) - a^2 y_{0_i}(t)$$
$$\dot{y}_{1_i}(t) = y_{4_i}(t)$$
$$\dot{y}_{4_i}(t) = Aa\{p_i(t) + \mathcal{K}\sum_{j=1}^{N} M_{ij}\mathcal{S}(y_{1_j}(t) - y_{2_j}(t)) \tag{6}$$
$$+ C_2\mathcal{S}[C_1 y_{0_i}(t)]\} - 2ay_{4_i}(t) - a^2 y_{1_i}(t)$$
$$\dot{y}_{2_i}(t) = y_{5_i}(t)$$
$$\dot{y}_{5_i}(t) = BbC_4\mathcal{S}(C_3 y_{0_i}) - 2by_{5_i}(t) - b^2 y_{2_i}(t)$$

where $M_{ij}$ is the adjacency matrix of the network and $\mathcal{K}$ represents the coupling coefficient between nodes. A population of pyramidal neurons interacts with one excitatory and one inhibitory population of neurons, as schematically shown in Fig 1(A). A network of nodes was formed by the connections between excitatory populations.

## 2.3 Network structure

Modeling biological, social, and many other real-world structures can be performed by networks of coupled dynamical systems, which include nodes (neurons or brain regions) and their biological connections. The matrix-based representation is a straightforward yet powerful method for illustrating the connectivity within brain networks. It provides a clear and concise way to map out the relationships between different neural components, facilitating a deeper understanding of the brain's structure. This method is particularly useful in the context of network neuroscience, a rapidly evolving field that applies graph theory and other mathematical tools to study the complex systems. A fundamental tool in matrix representation is the

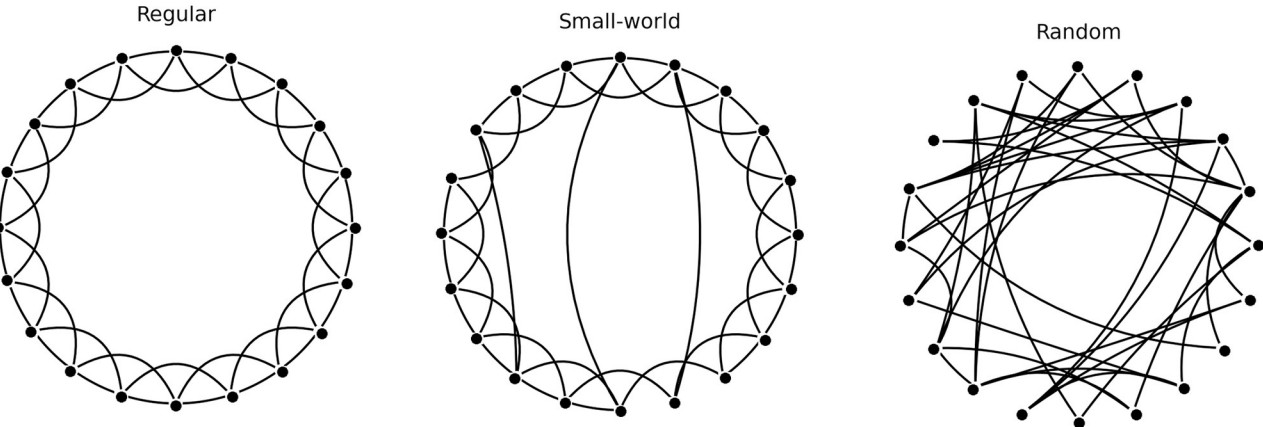

**Fig 2. Three Watts-Strogatz networks.** Each networks consisting of 20 nodes with the same number of 4 nearest neighbors. The probability of rewiring is 0 (Regular network), 0.2 (Small-world network), and 1 (Random network).

adjacency matrix, denoted as $M$, shows connections between nodes. Specifically, $M_{ij} = 1$ indicates a connection exists between node $i$ and node $j$, whereas $M_{ij} = 0$ signifies no connection.

Although a regular or random connection topology is usually assumed in most cases, many real systems follow a different topology. Indeed, real networks are characterized by high clustering coefficients and small mean-shortest path lengths, which are observed in regular and random structures, respectively [67]. These networks with these two properties are known as small-world networks. The Watts-Strogatz topology is capable of generating graphs with small-world properties [68]. A Watts-Strogatz topology is characterized by $N$ (number of nodes in a ring), $K$ (number of nearest neighbors of a node, K/2 on either side), and $p$ (the probability of rewiring assigned to each edge). The appropriate choice of these three parameters has a substantial role to build a Watts-Strogatz model. It is best to choose K such that the resulting network is neither sparse nor fully connected. Moreover, the following equation should hold: $N \gg K \gg \ln N \gg 1$. $p$ is the most challenging parameter to be estimated. This parameter plays a pivotal role in determining the network's small-world properties, as it influences the degree of randomness introduced into the network. At $p = 0$, the network is a regular lattice, while at $p = 1$, it becomes a random graph. $0 < p < 1$ result in networks that exhibit small-world characteristics, combining the benefits of both the lattice and random graph models. This flexibility allows for the exploration of how the small-world effect can be tuned by adjusting the rewiring probability, offering a powerful tool for studying the impact of network structure on various phenomena, from social networks to brain connectivity. Small-world properties are found in networks formed by $\frac{1}{N} \ll p \ll 1$ [69]. Fig 2 shows three Watts-Strogatz networks, each consisting of 20 nodes with the same number of 4 nearest neighbors. The probability of rewiring is 0 (Regular network), 0.2 (Small-world network), and 1 (Random network).

## 2.4 Kuramoto order parameter

The Kuramoto model consists of interacting oscillators that each of them is considered to have its own intrinsic natural frequency $\omega$, and each is coupled equally to all other oscillators. This

model is most commonly governed by the following equations:

$$\frac{d\phi_i}{dt} = \omega_i + \frac{1}{N}\sum_{l=1}^{N}D_{il}\sin(\phi_l - \phi_i), \quad i = 1, \dots, N \tag{7}$$

where $\phi_i$ is instantaneous phase of the $i$'s node, $D_{il}$ is the phase coupling matrix, and $N$ represents the number of nodes.

One way to recognize the degree of global synchronization in a network of coupled ensembles of identical oscillators is via the so-called Kuramoto order parameter (KOP). The value obtained with this criterion is equal to:

$$R(t) = \frac{1}{N}\left|\sum_{k=1}^{N}e^{j\phi_k(t)}\right| \tag{8}$$

where $\phi_k(t)$ followed the dynamics in Eq (7) and $j = \sqrt{-1}$. $R$ gives a value range in [0, 1] with 0 representing no phase synchrony (asynchronous state) and $R = 1$ when full synchronization occurs. Notably, the phase calculated in the KOP can be the result of the Hilbert transform of the oscillating signals [70, 71].

## 2.5 Measure of criticality

The study of phase transitions has been investigated over many decades in a broad range of physical systems and it is one of the most active research areas in statistical mechanics [72]. A system with a second-order phase transition (SOPT) or continuous phase transition sits at the transition point between two phases and separate an ordered and disordered one [73]. SOPT or criticality theory has sparked much interest over the years, as it has been demonstrated that critical behavior has many potential advantages such as maximal dynamical range, extensive information processing, and storage capabilities [74–76]. It is claimed that the healthy brain acts in a critical regime at a boundary between different types of dynamics [48, 54–56]. It is a challenging question that how critical dynamics can be detected in neural models. There are some measures of criticality that can be assessed [77–79]. Power-law scaling of neuronal avalanches [48], Branching ratio [80], operating at the edge of a phase transition [81], and medium mean with high variability in synchrony value [82] are some hallmarks of criticality that have been observed in neural models. In order for any marker of criticality to exist, first a critical point needs to be determined.

To investigate the SOPT, we compute the coefficient of variation (CV) of synchronization value against the control parameter (coupling coefficient). The sharp increase in the CV of the synchronization value across the coupling coefficient at the critical point is a key indicator of critical slowing down [83]. This phenomenon is particularly relevant in the study of SOPT and the behavior of complex systems near their critical points [84]. Indeed, the CV is a measure of the relative variability of a data series around its mean, and its increase at the critical point suggests that the variability of the synchronization value is significantly affected by the coupling coefficient at this point. This increase in CV can be interpreted as a sign that the system behavior is becoming more sensitive to changes in the coupling coefficient, indicating that the system is nearing a critical point and its dynamics undergo a qualitative change. Consequently, the peak in the CV during the continuous phase transition is a marker of criticality [85].

It is documented that normal brain function is characterized by long-range temporal correlations (LRTCs) in the cortex [52, 53]. As well as this, the critical hypothesis is supported by the presence of LRTCs in neural oscillation amplitudes [86]. Veritably, LRTCs are crucial characteristic of criticality. Detrended Fluctuation Analysis (DFA) or mean auto-correlation

methods are employed to investigate the temporal correlation structures of the signal. It is possible to detect LRTCs in a signal if its auto-correlation decays as a power-law with an exponent between −1 and 0 [87]. Generally, auto-correlation functions are very noisy in their tails, making exponent estimation difficult. In order to overcome these problems, DFA is an appropriate technique [53]. DFA determines the long-range temporal correlations in neuronal oscillation amplitude envelopes. The DFA method can be explained as follows: first, a time series $X(t)$ with a length of $N$ and $t \in 1, \ldots, N$, is divided into $N_T = \left(\frac{N}{T}\right)$ non-overlapping segments $X_i(t)$ with the same size $T$. A signal segment will be missed if $(N/T)$ is not an integer. We divide signals from their end to overcome this problem. So, we have $2N_T$ segments. This situation arises when the total length of the signal ($N$) is not exactly divisible by the segment length ($T$), leading to a remainder that cannot be accommodated within the defined segment length. This can result in the last segment being shorter than the others, potentially missing valuable data or introducing inaccuracies in the analysis. To address this issue, a common strategy is to divide the signal from its end, ensuring that each segment is of equal length and that no data is lost. This approach involves creating additional segments at the beginning of the signal, effectively padding the signal with extra data points. By doing so, the total number of segments becomes $2N_T$, where $N$ is the original signal length and $T$ is the segment length. This method ensures that every segment is of equal length, maintaining consistency and integrity in the analysis or processing of the signal. Then, the linear trend in segment $i$ is taken out, and the fluctuations $F_i(2N_T)$ corresponding to window length $2N_T$ are given as follows [88]:

$$F_i(2N_T) = \sqrt{\frac{1}{2N_T}\sum_{t=1}^{2N_T}(X_i(t) - X_i^{trend}(t))^2} \qquad (9)$$

where $X_i^{trend}(t)$ shows the linear trend in segment $i$. $F_i(2N_T)$ is plotted against $(2N_T)$ in a logarithmic scale. In power-law relation, $F_i(2N_T) \sim (2N_T)^{\alpha}$ which $\alpha$ is determined as the slope of a straight line fit, and $0.5<\alpha<1$ represents long-range correlation.

### 2.6 Simulation

We construct a network of 80 identical coupled neural phase oscillators as nodes to better adapt to a real network. The connections between units follow the small-world topology (Watts-Strogatz network with a rewiring probability of 0.2). Each node is connected to 20 neighbors, ten on each side. Then, WC and JR dynamics were applied to each node individually. Different initial conditions for both models are adjusted to show oscillatory activity in each independent run. The initial conditions refers to the initial value of varibles ($E$ and $I$ in WC model and ($y_0, \ldots, y_5$) for JR model) and external inputs in both models. These initial conditions change during repetitions, which can enhance precision and help mitigate bias in measurements. The length of this simulation is 200 s, and the time step is set to $10^{-4}$ s. For every coupling strength, each simulation is repeated twenty times. In this study, the network is analyzed by changing the global coupling strength $\mathcal{K}$.

### 3 Results

We consider the output series of each node and calculate the phase synchronization by KOP measure between them at each trial. First, a network with no copupling is considered. Then, we increased copuplig coefficient until each node leaves the limit cycle and rests in a fixed point. Fig 3 shows the mean synchrony value against the coupling coefficient in JR (**A**) and WC (**B**) models. This value refers to the average degree of synchronization among the oscillators in the network. Vertical blue bars represent dispersions for twenty runs (standard

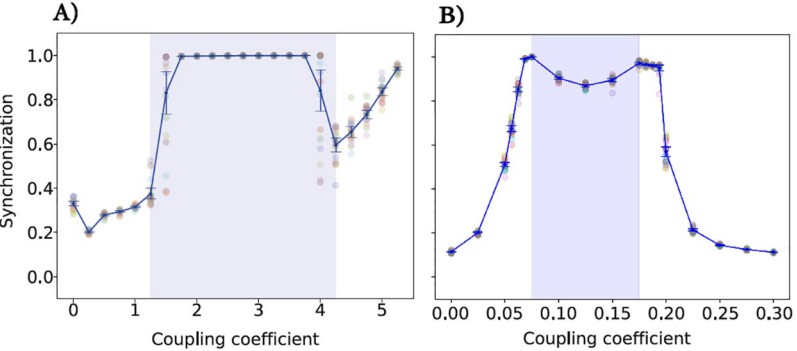

**Fig 3. The KOP value according to the coupling coefficient in the JR (A) and WC (B) model.** (A) At $\mathcal{K} = 1.25$ and 4.25, the dispersion of the synchronization value is high, and the area between them is related to the high synchrony regime. (B) At $\mathcal{K} = 0.075$ and 0.175, the blue highlighted regime shows an unexpected (decreasing in the synchrony value) pattern. The simulation runs twenty times on each coupling coefficient for these two models.

deviations). This provides a measure of the variability in the results: a larger standard deviation means the results were more spread out and less consistent. According to the results in Fig 2, depending on the coupling coefficient, different synchronization values arise from collective behaviors. This means that the degree to which the oscillators in the network are synchronized can change depending on how strongly they influence each other. For example, if the coupling coefficient is high, the oscillators may become completely synchronized, leading to a state where all oscillators behave the same way. On the other hand, if the coupling coefficient is low, the oscillators may behave more independently and the network may be less synchronized. Both models exhibited a transition from an unsynchronized to a highly synchronized state. This kind of behavior was detected in ensembles of coupled oscillators [61, 89–91]. A similar occurrence of transition from unsynchronized to full phase synchronized state is observed in [79].

As shown in Fig 3(A) increasing coupling up to 1.25 causes more synchronization. When the coupling coefficient is set to 1.25, the network's synchronization significantly increases, indicating a potential disorder or instability in the network's function [92–94]. Our results indicated that in the [1.25, 4.25] regime of coupling strength, the synchronization values are high. We call this area a high synchrony regime and the coupling coefficient equal to 1.25 (4.25) is the starting (ending) point of high synchrony behavior. This suggests that these are the values of the coupling coefficient at which the network transitions between high and low synchronization. The coupling coefficient of 1.5 and 4 produces a large variation in the mean of the KOP, accompanied by a large variance. These values can serve as potential phase transition points, which are points where the network transits from one state to another. In this case, the transition could be from a state of lower synchronization to a state of higher synchronization, or vice versa.

In Fig 3(B), between $\mathcal{K} = 0.075$ to 0.175, the decrease in the synchrony value is seen. Indeed, Fig 3(B) shows a local minimum in the KOP in which by increasing coupling strength, the synchronization value decreases. It appears that this type of behavior also occurs in a network of Kuramoto model [95]. Now, to investigate the partial synchronization, we plot the colored output matrix of JR and WC network activity in Figs 4 and 5, respectively. In a range of small coupling coefficients, the system activities are randomly distributed, related to incoherent states. In Fig 4 the network shows the complete synchronization for $2 < \mathcal{K} < 4$ which corresponds to coherent states. This means that the network becomes highly coordinated or synchronized

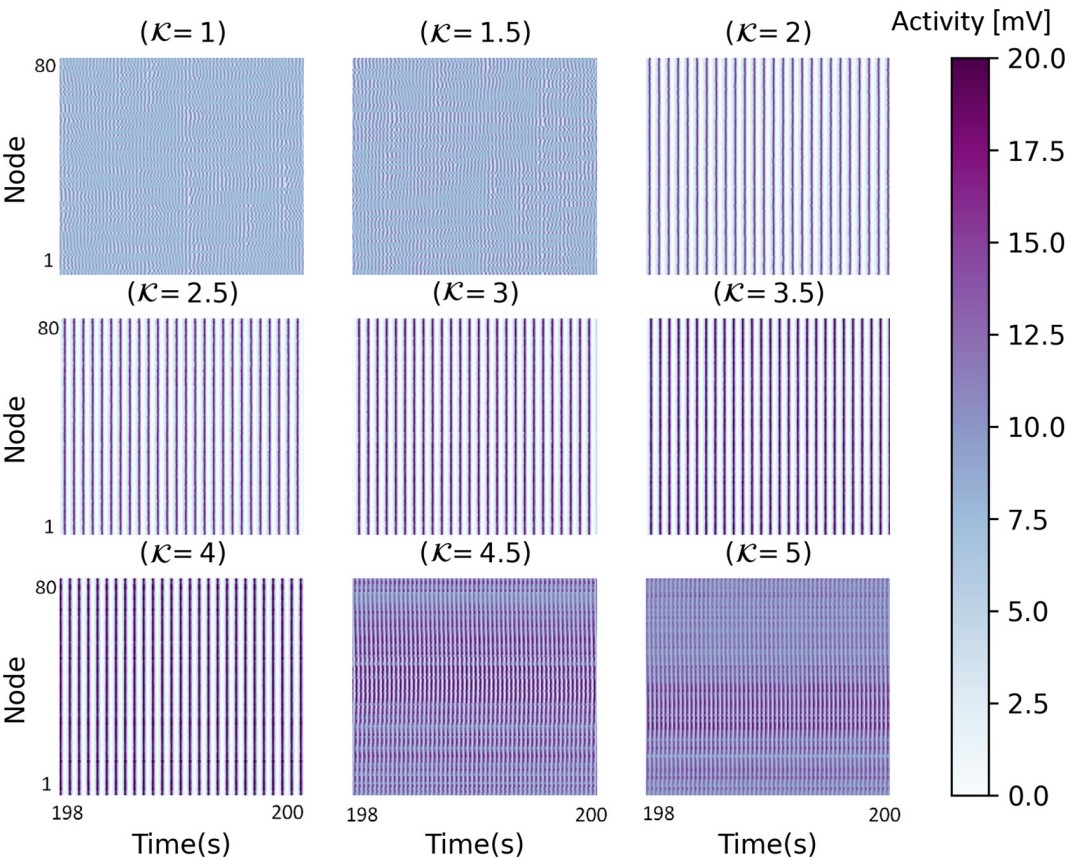

**Fig 4. The colored output matrices during the last two seconds for different values of coupling coefficient in the JR model.** In $\mathcal{K}$ = 1.25 to 4.25, the output signal shows an ordered behavior.

when the coupling strength is between 2 and 4. The nervous system in this condition is not in a healthy state and may lead to some brain disorders such as epileptic seizure activity [92, 96, 97]. It would seem that high levels of synchronization in a neuronal network could be the origin of abnormal brain activities. Indeed, the system activity is coherent in space and time and shows global synchronization, relating to its highly synchronized nature. In Fig 5, the colored activity matrix of the output signal shows a slight time delay synchronization corresponding to the blue region in Fig 3(B) ($0.075 < \mathcal{K} < 0.175$). It means that the oscillators are not necessarily in phase (i.e., they do not oscillate simultaneously), but there is a consistent delay between the different oscillators. Herewith the rate of the excitatory ensemble of neurons in WC model is coherent in time and incoherent in space which means that while the neurons fire in a coordinated manner over time, they do not necessarily fire at the same locations in the network. This could be due to the spatial distribution of the neurons in the network, or it could be a property of the neurons themselves. Indeed, the nodes are synchronized with delay to generate some patterns of partial synchronization that are evident only in this model and not in JR one. It is a confirmation that different synchronization behaviors may result from different models of neural networks. A large number of neurologic disorders, including schizophrenia, Alzheimer's disease, and Parkinson's, may be associated with the coexistence of synchronous or asynchronous [16, 98, 99]. This finding is in line with the increasingly recognized role of neural synchrony in various brain functions and disorders.

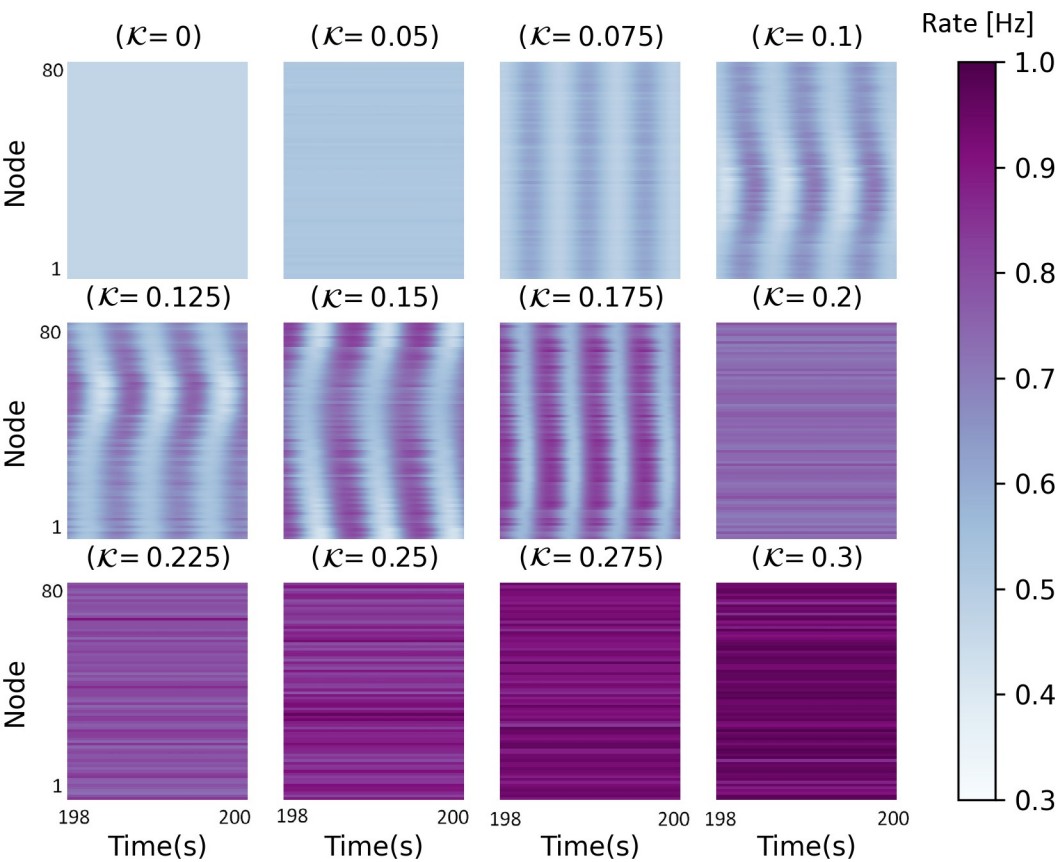

**Fig 5. The colored output matrices during the last two seconds for different values of coupling coefficient in the WC model.** In $\mathcal{K}$ = 0.075 to 0.175, the output signal shows a slight time delay synchronization and the nodes are synchronized with a slight delay.

In order to analyze the SOPT, we compute the CV of Fig 3 against the control parameter (coupling coefficient) in Fig 6. Two peaks in the CV during the continuous phase transition (at $\mathcal{K} = 1.25, 4.25$) are markers of criticality (Fig 6(A)) in JR model. These points of maxima of this curve correspond to the value of the transition points in Fig 3(A), indicating that the

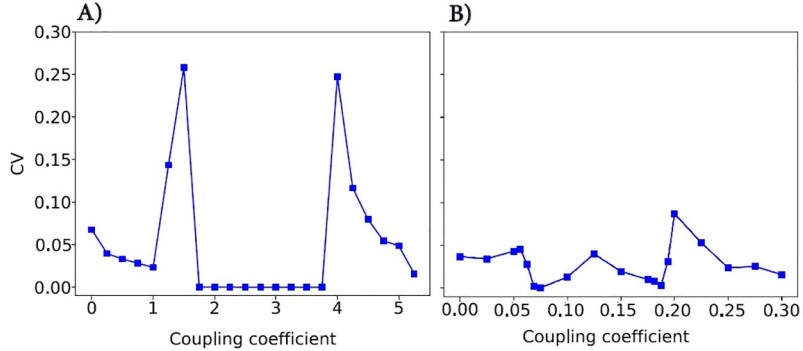

**Fig 6. CV vs. coupling coefficient in (A) JR and (B) WC.** The maxima in (A) correspond to transition points observed in Fig 3.

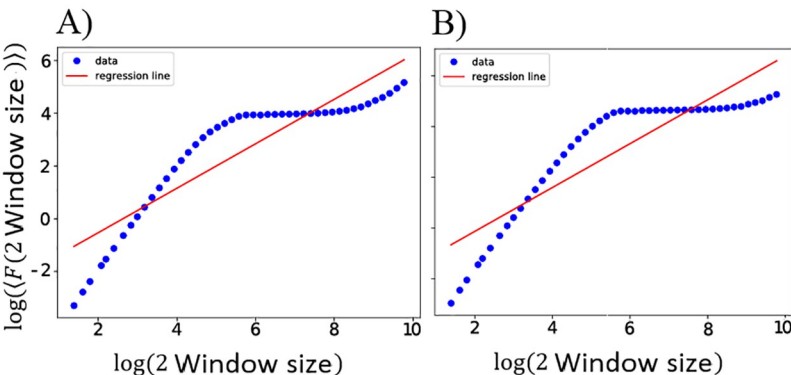

**Fig 7. The fluctuation plot for an arbitrary value in JR (A) and WC (B) models in the log–log plot.** The values on the x-axis are in seconds on logarithmic scales based on segment sizes, and the y-axis shows the standard deviation mean of all sized segments. This trend of data is piecewise linear, and the linear fit is not suitable for them. LRTCs do not exist at these points.

system transits from one state to another at these coupling coefficients. Fig 6(B) does not show any peak, which means WC model does not exhibit criticality. This suggests that the WC model does not undergo a continuous phase transition at the same coupling coefficients as the JR model. This could indicate that different models of neural networks may exhibit different dynamical behaviors. Now, we examine LRTC in these two models. LRTCs in the amplitude of neural oscillations support the critical dynamics of biological neuronal networks. Fig 7 shows an arbitrary realization of the applied DFA technique in an absolute signal Hilbert transformation with no overlapping. Based on these results, it appears that the linear fit does not fit the data in either case. Hence, LRTCs are not present at these points. A piece-wise linear function (purple line) is suitable for fitting data which confirms the deviation from linearity. It appears that the system's behavior does not follow a simple linear trend and may involve more complex dynamics. Similarly, the presence of LRTCs has not been observed in other coupling coefficients, so this feature is not apparent in either of these models with defined parameters. This could mean that the system's dynamics is not close to a critical state. These results confirm previous findings [29, 79].

## 4 Discussion and conclusion

In this study, we compared the dynamics of two seminal neural mass models, representing the integration of many neurons within a population. In the past few decades, neural mass models have contributed significantly to our understanding of meso- and macroscale dynamics of neuronal populations [61, 66, 89]. A wide range of neural phenomena has been modeled using them, such as alpha band oscillations [100], sleep rhythms [4], evoked potentials [8], and mechanisms of brain rhythms in disease [5, 101].

We investigated phase synchronization and criticality in two voltage-based (JR) and rate-based (WC) neural mass models. The two models were utilized in the same structural network representing the same topology. We observed two profound phenomena in brain activity. First, the global synchronicity of these networks was calculated by considering the KOP. The phase synchronization analysis of WC and JR neural mass models revealed that they shared a common feature. There was a transition from low to full synchronization in both models, which is not indicative of a healthy system, however, JR was the only model that showed a phase transition. This is likely because, in the WC model, the dynamics of a mass is expressed

by a two-dimensional system. Indeed, WC model based on the equilibrium of both excitatory and inhibitory inputs shows only the fixed-points and oscillatory activities and may not describe the biophysical details of brain activity [10]. This model has revealed the presence of Hopf bifurcations [102], but JR dynamics is described by a set of six ordinary differential equations and can differentiate between different brain waves. A single JR mass can produce different types of activities including epileptic spike-like activity and alpha-like ones that do not appear in WC model. Moreover, the detailed bifurcation analysis of JR has shown an impressively rich bifurcation diagram [101].

Another measure that we studied in these two networks is partial synchronization. Partial synchronization in neural models have attracted the attention of researchers. The emergence of different patterns in partial synchronization supports both healthy and disordered states in brain functions. There is some evidence that networks of Wilson-Cowan oscillators can exhibit a wide range of nonlinear behaviors [103]. Furthermore, our comprehension of the spatiotemporal dynamics of electrical activity, which have more similarities with functional networks, has been increased through the analysis of WC networks [104]. In this work, we observed that WC model could not show the second-phase transition, however, does not exclude complex dynamics. This model generally synchronizes its nodes with a small time delay. Indeed, two synchronous or asynchronous behaviors appeared simultaneously in this network, which was not seen in JR model.

Moreover, we investigated power-law behavior. Our results showed that even though JR model demonstrated a phase transition under the defined conditions, none of these models were capable of exhibiting power-law behavior and consequently criticality. Indeed, the occurrence of a continuous phase transition at a critical point is a necessary but not sufficient condition. Critical points exhibit clear characteristics and invariance of scale behavior. These phase transition points do not demonstrate scaling invariance, therefore they are not critical.

In summary, neural mass models for modeling brain dynamics depend on several factors, including the specific characteristics of the neural activity being modeled, the level of detail required, and the computational resources available. Both models have their strengths and are suited to different scenarios. JR model is more suitable for detailed neural dynamics and it provides a comprehensive framework for understanding the complex interactions within neural populations. Conversely, the WC focuses on the interaction between excitatory and inhibitory populations, providing a more straightforward representation of neural dynamics. While it may not capture the full complexity of neural interactions as the JR does, it is better suited for broader applications where a simplified yet accurate representation of neural dynamics is sufficient. The choice between these models should be guided by the specific requirements of the modeling task, including the level of detail needed, the presence of stochastic processes, and computational constraints.

It is notable that different parameter sets and different topologies have not been discussed, which implies the limitation of our research. These results do not apply to the WC or JR models in general, but with a specific parameter set and on a specific network topology. Different topologies (changes in the number of nearest neighbors of a node or in the probability of rewiring), or different parameter sets, are likely to exhibit entirely different behaviors. Moreover, we emphasis that these analysis does not exclude that transitions could be generated also by varying other model parameters, or with different topologies.

We are left with the question of whether any of these models can be modified so that they exhibit power-law and partial synchronization. Additionally, discovering the optimal neural mass model for investigating brain dynamics can be challenging. This question arises from the nature of the specific features selected to observe these findings. We also performed power spectrum analysis, bifurcation analysis, and entropy analysis of both networks, which were not

included in the results section because they did not affect our understandings from the phase transition analysis. Indeed, it is a key to search for features that show a higher level of power analysis. As a final note, it is unclear how a different type of partial synchronization, such as cluster synchrony appear in these types of networks, which can be another research subject for future studies.

## Acknowledgments

The authors would like to thank the anonymous reviewers for their valuable comments and suggestions.

## Author Contributions

**Conceptualization:** Sheida Kazemi, Yousef Jamali.

**Formal analysis:** Sheida Kazemi.

**Investigation:** Sheida Kazemi, Yousef Jamali.

**Resources:** Yousef Jamali.

**Supervision:** Yousef Jamali.

**Validation:** Sheida Kazemi, AmirAli Farokhniaee, Yousef Jamali.

**Visualization:** Sheida Kazemi, AmirAli Farokhniaee.

**Writing – original draft:** Sheida Kazemi, AmirAli Farokhniaee.

**Writing – review & editing:** Sheida Kazemi, AmirAli Farokhniaee.

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
