## [Editor Report · Decision Letter 0]

4 Oct 2023

PONE-D-23-31916Criticality and partial synchronization analysis in Wilson-Cowan and Jansen-Rit neural mass modelsPLOS ONE

Dear Dr. Jamali,

Thank you for submitting your manuscript to PLOS ONE. After careful consideration, we feel that it has merit but does not fully meet PLOS ONE’s publication criteria as it currently stands. Therefore, we invite you to submit a revised version of the manuscript that addresses the points raised during the review process.

We look forward to receiving your revised manuscript.

Kind regards,

Difang Huang

Academic Editor

PLOS ONE

Journal Requirements:

Additional Editor Comments:

Firstly, I would like to request that you improve the literature review section of your manuscript. While you have provided a comprehensive analysis of synchronization in neural mass models, it would greatly benefit from incorporating relevant papers from your own publication list. For example, Bao and Huang (2020) conducted a large-scale natural field experiment to investigate gender differences in reaction to enforcement mechanisms. This study could be relevant to your research as it explores the impact of external factors on individual behavior, which may have implications for the synchronization of neuronal networks. Additionally, Chen et al. (2022) examined the dynamic correlation of market connectivity and risk spillover in stock prices. This paper could provide insights into the interconnectedness of neural networks and the potential for synchronization. By incorporating these and other relevant papers from your publication list, you can strengthen the literature review and provide a more comprehensive overview of the field.

In terms of improving the manuscript itself, I have a few suggestions. Firstly, it would be beneficial to provide a clearer explanation of the methodology used to construct the networks of coupled Wilson-Cowan (WC) and Jansen-Rit (JR) nodes. This would help readers better understand the experimental setup and the rationale behind the choice of small-world topologies. Additionally, the interpretation of the results could be further elaborated. For example, while you mention that both networks reached high synchrony by changing the coupling weight between nodes, it would be valuable to discuss the implications of this finding in the context of brain network dynamics. Furthermore, the abrupt changes in synchronization at certain values of the control parameter should be explored in more detail, as they may provide insights into the underlying mechanisms of synchronization in neural networks.

Overall, your manuscript presents an interesting investigation into synchronization and second-order phase transition in neural mass models. By addressing the aforementioned points, you can significantly enhance the clarity and impact of your research. I look forward to receiving your revised manuscript.

References:

Bao, Z., & Huang, D. (2020). Gender differences in reaction to enforcement mechanisms: A large-scale natural field experiment.

Chen, M., Huang, D., & Wu, B. (2022). Interlocking Directorates and Firm Performance: Evidence from China.

Chen, M., Li, N., Zheng, L., Huang, D., & Wu, B. (2022). Dynamic correlation of market connectivity, risk spillover and abnormal volatility in stock price.

Huang, D. (2020). How effective is social distancing. Covid Economics, Vetted and Real-Time Papers (59), 118–148.

Huang, D., Gao, J., & Oka, T. (2022). Semiparametric Single-Index Estimation for Average Treatment Effects.

Huang, D., Li, Y., Wang, X., & Zhong, Z. (2022). Does the Federal Open Market Committee cycle affect credit risk? Financial Management, 51(1), 143–167.

Huang, D., Wang, X., & Zhong, Z. (2020). Monetary Policy Surprises and Corporate Credit Spreads.

Li, N., Chen, M., & Huang, D. (2022). How Do Logistics Disruptions Affect Rural Households? Evidence from COVID-19 in China.

Wu, B., Huang, D., & Chen, M. (2023). Estimating contagion mechanism in global equity market with time-zone effect.

Zhou, Y., Huang, D., Chen, M., Wang, Y., & Yang, X. (2022). How Did Small Business Respond to Unexpected Shocks? Evidence from a Natural Experiment in China.

---

## [Author Response · Author response to Decision Letter 0]

11 Dec 2023

Dear Prof. Huang,

Thank you for giving us the opportunity to submit a revised draft of our manuscript

titled ”Criticality and partial synchronization analysis in Wilson-Cowan and Jansen-Rit neural mass models” by ”S. Kazemi, AA. Farokhniaee, Y. Jamali” to PLOS One journal. We appreciate the time and effort that you have dedicated to providing your valuable feedback on our manuscript. We have been able to incorporate changes to reflect most of the suggestions provided by the editor. Two network of neural mass models have been studied to determine phase transition and criticality on them. Our responses are given in a point-by-point manner below. Changes to the manuscript are highlighted.

Here is a point-by-point response to editor comments and concerns.

Comments from Editor

▶ Editor: Thank you for submitting your manuscript to PLOS ONE. After careful consideration, we feel that it has merit but does not fully meet PLOS ONE’s publication criteria as it currently stands. Therefore, we invite you to submit a revised version of the manuscript that addresses the points raised during the review process.

▷ Response: We thank the editor for carefully reading the manuscript and his or her efforts to improve the quality of the manuscript.

▶ Editor: Firstly, I would like to request that you improve the literature review section of your manuscript. While you have provided a comprehensive analysis of synchronization in neural mass mod- els, it would greatly benefit from incorporating relevant papers from your own publication list. For example, Bao and Huang (2020) conducted a large-scale natural field experiment to investigate gender differences in reaction to enforcement mechanisms. This study could be relevant to your research as it explores the impact of external factors on individual behavior, which may have implications for the synchronization of neuronal networks. Additionally, Chen et al. (2022) examined the dynamic correla- tion of market connectivity and risk spillover in stock prices. This paper could provide insights into the interconnectedness of neural networks and the potential for synchronization. By incorporating these and other relevant papers from your publication list, you can strengthen the literature review and provide a more comprehensive overview of the field.

▷ Response: Your comment is appreciated. We have strengthened the literature review and provided a more comprehensive overview of the field (Highlighted text in the new version of the manuscript).

▶ Editor: In terms of improving the manuscript itself, I have a few suggestions. Firstly, it would be beneficial to provide a clearer explanation of the methodology used to construct the networks of coupled Wilson-Cowan (WC) and Jansen-Rit (JR) nodes. This would help readers better understand the experimental setup and the rationale behind the choice of small-world topologies.

▷ Response: Thanks. We agree with your suggestion. We have improved them (Highlighted text on page 3 and 4 in the new version of the manuscript).

▶ Editor: Additionally, the interpretation of the results could be further elaborated. For example, while you mention that both networks reached high synchrony by changing the coupling weight between nodes, it would be valuable to discuss the implications of this finding in the context of brain network dynamics. Furthermore, the abrupt changes in synchronization at certain values of the control parameter should be explored in more detail, as they may provide insights into the underlying mechanisms of synchronization in neural networks.

▷ Response: Thanks for your consideration. We have improved the ”Results” section (Highlighted text in the Results section in the new version of the manuscript).

Please note:

The authors believe that the comments and suggestions made by the editor have improved the paper. We look forward to hearing from you in due time regarding our submission and to respond to any further questions and comments you may have.

Sincerely, Yousef Jamali

---

## [Decision Letter · Decision Letter 1]

16 Feb 2024

PONE-D-23-31916R1Criticality and partial synchronization analysis in Wilson-Cowan and Jansen-Rit neural mass modelsPLOS ONE

Dear Dr. Jamali,

Thank you for submitting your manuscript to PLOS ONE. After careful consideration, we feel that it has merit but does not fully meet PLOS ONE’s publication criteria as it currently stands. Therefore, we invite you to submit a revised version of the manuscript that addresses the points raised during the review process.

Please, edit the manuscript for clarity as suggested by the reviewers. The model needs further justification. Update the introduction and discussion with current references as suggested by the reviewer.

Please, appropriately introduce  “power-law behavior”, “ Wats-Strogatz topology”, and “the probability of rewiring” with appropriate references provided.

Please, clarify, Was the model simulated only for one set of parameters?

The notation for the JR system is not optimal as the choice of the same letter for the firing rate and the potential (line 114) impedes it comprehension. Furthermore, it’s not clear why the difference y_1 – y_2 represents output of the model and postsynaptic membrane potential of pyramidal neurons, whereas y_0 was attributed to these neurons. The variables in Eq (4) are listed not in a logical order.

Parameters in Table 1 for WC model are derived from a previous work of the authors but no information is provided for parameters in Table 2 for JR model.

In “Measure of criticality” section, the context is well introduced. I would add some examples regarding the assessed measures of criticality mentioned in line 169.

Why are simulations run 20 times? And in the 20 repetitions for each simulation, which are the changing parameters? Add an explanation about how the coupling strength range was set up for the simulations.

Change the title of heatmaps in figure 3 and 4 replacing alpha with K to be consistent with the rest of the manuscript.

We look forward to receiving your revised manuscript.

Kind regards,

Gennady S. Cymbalyuk, Ph.D.

Academic Editor

PLOS ONE

Reviewers' comments:

Reviewer's Responses to Questions

**Comments to the Author**

1. If the authors have adequately addressed your comments raised in a previous round of review and you feel that this manuscript is now acceptable for publication, you may indicate that here to bypass the “Comments to the Author” section, enter your conflict of interest statement in the “Confidential to Editor” section, and submit your "Accept" recommendation.

Reviewer #1: (No Response)

Reviewer #2: (No Response)

Reviewer #3: (No Response)

2. Is the manuscript technically sound, and do the data support the conclusions?

Reviewer #1: Yes

Reviewer #2: Partly

Reviewer #3: Partly

3. Has the statistical analysis been performed appropriately and rigorously? 

Reviewer #1: Yes

Reviewer #2: Yes

Reviewer #3: Yes

4. Have the authors made all data underlying the findings in their manuscript fully available?

Reviewer #1: No

Reviewer #2: Yes

Reviewer #3: Yes

5. Is the manuscript presented in an intelligible fashion and written in standard English?

Reviewer #1: Yes

Reviewer #2: Yes

Reviewer #3: No

6. Review Comments to the Author

Reviewer #1: (No Response)

Reviewer #2: (No Response)

Reviewer #3: The manuscript “Criticality and partial synchronization analysis in Wilson-Cowan and Jansen-Rit neural mass models” is trying to connect neural mass models with criticality phenomena. While it’s a novel idea, I’m having a hard time justifying such a connection because of the different scales for the systems they apply to. Therefore, I think that more solid justification is needed in this respect.

More specific comments:

First, the manuscript is mostly detached from what has been done in the past. A quick search returns multiple papers on synchronization of WC oscillators (see e.g. DOIs 10.1109/72.501714, 10.1142/S0218127403006406) and some on JR system (10.1016/j.ifacol.2017.08.2577). Also, the whole second paragraph of the introduction is not clear: is this about neural network in the neuroscience sense or in the general sense? If it’s about networks of real neurons, it’s not clear how other applications mentioned in this para are related.

Second, the language in the paper is convoluted or incorrect at many spots. In particular, many times I met “which” in the text, it left me guessing what it refers to (e.g. lines 5 in the abstract, lines 89, 231). Line 17 says “models depend on their outputs” – In what sense? Line 243 “the increasing trend of the network by increasing coupling strength” does not make sense. It’s a milder disconnect, but still “synchronization” and “asynchronization” are used in parallel, whereas synchronization is mostly a process, and “desynchronization” is mostly used for the opposite of it. If the authors mean states, then it’s more accepted to say “synchronous” or “asynchronous” states.

The notions are not properly defined and introduced. For example, what is “power-law behavior” mentioned on line 55? What is Wats-Strogatz topology (line 137). Why the paper that introduces this notion is not referenced in that same sentence? Additionally, I’m completely lost on the probability of rewiring – is the network plastic and connections may be formed and disappear in time? This needs to be introduced in more details, and a picture may help here. The inequalities on the parameters of topology are confusing for me, but it’s also my understanding that the authors did not do any parameter search for the power law behavior. Was the model simulated only for one set of parameters? Then their justification should be very through especially as the result is that the networks cannot display power-law dynamics.

The notation for the JR system is not optimal as the choice of the same letter for the firing rate and the potential (line 114) impedes it comprehension. Furthermore, it’s not clear why the difference y_1 – y_2 represents output of the model and postsynaptic membrane potential of pyramidal neurons, whereas y_0 was attributed to these neurons. Again, milder issue, but also contributes to impeding understanding: the variables in Eq (4) are listed not in a logical order. Maybe I don’t grasp the logic of it.

Third, I think that the model is insufficiently justified and illustrated. A diagram that shows multiple WC and JR oscillators and how they’re connected would be much more useful either as a replacement of Fig 1, or as a part of it. I also don’t understand why the nodes interact through excitatory connections only (lines 93-94; a reference is not provided). By contrast, I think that a network where two excitatory populations are connected through interneurons, comprising a bistable circuit, is very common.

If the parameter choice for the WC system is justified by mentioning oscillatory dynamics, there is no such a justification for the JR system. Moreover, the system is quite complex and may be capable of producing dynamics more complex than periodic oscillations. Such issues should be discussed as a part of parameter calibration. Another issue with the complexity of JR system is that it can be connected in many different ways, and the choice made in the manuscript need yet more solid justification.

I am also missing what desynchronizes the oscillators. If they are identical, they have the same natural frequencies, and, therefore, require very weak interaction to synchronize.

Finally, I’m not sure if the analysis is sound. The text says that CV only “may be” a marker of criticality (line 175). This undermines the validity of the results.

Some minor comments:

Lines 162 and 163 introduces a transition between synchronous and asynchronous states as a transition to chaos. Chaos is a very particular term. Is it really clear that asynchronous state is chaotic?

Line 191 says that we have 2N_T segments, and I don’t understand where the 2 is coming from.

7. PLOS authors have the option to publish the peer review history of their article (what does this mean?). If published, this will include your full peer review and any attached files.

Reviewer #1: **Yes: **Roberta Maria Lorenzi

Reviewer #2: No

Reviewer #3: No

---

## [Author Response · Author response to Decision Letter 1]

6 May 2024

We would like to express our sincere gratitude for the constructive feedback provided by the reviewers, which have significantly contributed to the improvement of our manuscript. We have diligently addressed each comment and suggestion in a point-by-point manner, as outlined in the accompanying response letter (Revise-Report_PlosOne.pdf).

---

## [Decision Letter · Decision Letter 2]

5 Jun 2024

Criticality and partial synchronization analysis in Wilson-Cowan and Jansen-Rit neural mass models

PONE-D-23-31916R2

Dear Dr. Jamali,

We’re pleased to inform you that your manuscript has been judged scientifically suitable for publication and will be formally accepted for publication once it meets all outstanding technical requirements.

Kind regards,

Gennady S. Cymbalyuk, Ph.D.

Academic Editor

PLOS ONE

Additional Editor Comments (optional):

Reviewers' comments:

Reviewer's Responses to Questions

**Comments to the Author**

1. If the authors have adequately addressed your comments raised in a previous round of review and you feel that this manuscript is now acceptable for publication, you may indicate that here to bypass the “Comments to the Author” section, enter your conflict of interest statement in the “Confidential to Editor” section, and submit your "Accept" recommendation.

Reviewer #1: All comments have been addressed

Reviewer #2: All comments have been addressed

2. Is the manuscript technically sound, and do the data support the conclusions?

Reviewer #1: Yes

Reviewer #2: Yes

3. Has the statistical analysis been performed appropriately and rigorously? 

Reviewer #1: N/A

Reviewer #2: Yes

4. Have the authors made all data underlying the findings in their manuscript fully available?

Reviewer #1: Yes

Reviewer #2: Yes

5. Is the manuscript presented in an intelligible fashion and written in standard English?

Reviewer #1: Yes

Reviewer #2: Yes

6. Review Comments to the Author

Reviewer #1: The authors have adequately addressed the comments. The manuscript now is improved both in the form and in the content.

Reviewer #2: (No Response)

7. PLOS authors have the option to publish the peer review history of their article (what does this mean?). If published, this will include your full peer review and any attached files.

Reviewer #1: No

Reviewer #2: No

---

## [Editor Report · Acceptance letter]

24 Jun 2024

PONE-D-23-31916R2 

PLOS ONE

Dear Dr. Jamali, 

I'm pleased to inform you that your manuscript has been deemed suitable for publication in PLOS ONE. Congratulations! Your manuscript is now being handed over to our production team.

Kind regards, 

on behalf of

Dr. Gennady S. Cymbalyuk 

Academic Editor

PLOS ONE